# Wearable Neurotechnology for the Treatment of Insomnia: The Study Protocol of a Prospective, Placebo-Controlled, Double-Blind, Crossover Clinical Trial of a Transcranial Electrical Stimulation Device

**DOI:** 10.3390/clockssleep7010003

**Published:** 2025-01-26

**Authors:** Keenan Caswell, Grace Roe, Emamoke Odafe, Subodh Arora, Caddie Motoni, John Kent Werner

**Affiliations:** 1Department of Pediatrics, Walter Reed National Military Medical Center, Bethesda, MD 20814, USA; keenan.r.caswell@gmail.com; 2F. Edward Hébert School of Medicine, Uniformed Services University, Bethesda, MD 20814, USA; 3Sleep Disorders Center, Walter Reed National Military Medical Center, Bethesda, MD 20814, USA; 4Department of Neurology, Walter Reed National Military Medical Center, Bethesda, MD 20814, USA

**Keywords:** transcranial electrical stimulation, insomnia, sleep onset latency, actigraphy, electroencephalogram

## Abstract

Sleep disruption and deprivation are epidemic problems in the United States, even among those without a clinically diagnosed sleep disorder. Military service members demonstrate an increased risk of insomnia, which doubles after deployment. This study will investigate the ability of a translational device, Teledyne PeakSleep™ (Teledyne Scientific & Imaging, Durham, NC, USA), to reduce sleep onset latency and the time spent awake after sleep onset, with improvement in the subjective benefits of sleep for patients with insomnia by enhancing the brain rhythms within the frontal lobe implicated in slow wave generation. During this crossover trial, patients will use the wearable neurotechnology prototype headband, which delivers < 14 min of frontal short duration repetitive–transcranial electrical stimulation over a 30 min period immediately before trying to fall asleep. Using active stimulation versus a sham paradigm, we will compare actigraphy data, physiological data, and subjective sleep measures against a pre-treatment baseline in the same patient over the course of the 8-week study. If successful, PeakSleep™ could address the final common pathway in insomnia, namely the onset and maintenance of slow-wave sleep (SWS), and accordingly has the potential to enhance sleep onset in a wide range of individuals, most importantly warfighters in whom efficient sleep onset may be critical for operational success.

## 1. Introduction

Insufficient sleep syndrome, often with comorbid insomnia, has become a health problem of epidemic proportions within the United States military. Over 50% of warfighters chronically sleep less than 6 h per night while on deployment or in their garrison. Consequently, they are predisposed to workplace injury, cognitive and physical health compromises, poor morale, a weakened immune system, and fatal/costly accidents [1,2,3,4]. One-half to two-thirds of the 2.5 million U.S. military troops that have served in Afghanistan and Iraq have reported insomnia after returning home [5,6,7]. Issues with insomnia persist in veterans and frequently worsen with age, creating a huge burden on the Veterans Affairs (VA) healthcare system [8].

In addition to our veteran and active-duty population, sleep disruption and deprivation remain an epidemic problem in the United States, even among those without a clinically diagnosed sleep disorder [9,10]. There are a large number of available pharmacological sleep aids and cognitive behavioral therapies for healthy individuals, but each comes with recurring costs, limited access, short-term efficacy, and potentially undesirable side effects [2,11,12]. By comparison, transcranial electrical stimulation (tES) has been demonstrated to be safe across thousands of users and carries a favorable side effect profile, with the primary side effect of low-dose tES being a tingling, prickly, or itchy sensation at the electrode site [13,14]. tES has been shown to enhance deep sleep and memory consolidation when applied during a nap or overnight sleep [15,16,17,18,19]. Recently, several studies and ongoing trials have investigated whether tES applied prior to sleep or during nighttime awakenings can achieve similar benefits to tES during sleep by reducing sleep onset latency (SOL), increasing sleepiness, or improving subjective sleep quality [20,21,22,23,24,25].

Recognizing the clinical and operational needs for improving sleep onset and quality, Teledyne developed a field-ready, wearable neurotechnology device for sleep modulation called PeakSleep™. The device consists of a soft fabric headband with integrated electronics for dry electroencephalogram (EEG) sensing and electrical stimulation (see Figure 1). Teledyne internally investigated tES with a phase I study in 2019–2020 in 23 civilian participants with sleep onset insomnia who were randomized in a double-blinded, sham-controlled crossover study. The trial replicated previously reported work demonstrating that applying tES (0.75 Hz) over the forehead prior to sleep onset can enhance inter-stimulus delta power. Teledyne’s study revealed a reduction in SOL with a remarkably large effect size for a trial with only 23 participants (Cohen’s d = 1.3) (manuscript in preparation). Furthermore, patients reported an overall sleep benefit and improved sleep quality on a Likert scale. These results provided the first-in-human clinical evidence of a safe, robust, and portable sleep priming device.

Building on this trial, Teledyne Scientific & Imaging, Limited Liability Company, (LLC) and the Uniformed Services University of the Health Sciences (USU) have partnered together to further investigate the device’s ability to reduce SOL and wakefulness after sleep onset (WASO) while improving restfulness and subjective sleep for patients with insomnia. We hypothesize that by applying tES across the forehead/mastoid, we can impose slow oscillations within the cortex during wakefulness prior to sleep, and the brain will be more likely to enter a sleep state. We aim to develop and demonstrate a new treatment option for patients with insomnia that is effective, safe, and poses a favorable side effect profile.

## 2. Experimental Design

### 2.1. Study Design

The tES study (Wearable Neurotechnology for the Treatment of Insomnia) is a prospective, placebo-controlled, double-blind, crossover phase II clinical trial of a non-significant risk device used to treat individuals with sleep-onset insomnia. This study will be run as a single investigational site study with Teledyne Scientific & Imaging, LLC (Durham, NC, USA), as the Sponsor, and the Uniformed Services University of the Health Sciences (USU) as the trial investigation site. The study will involve five in-person visits to the study site, each separated by two weeks, for eight weeks total. Participants will be randomly assigned to either the short-duration repetitive–transcranial electrical stimulation (SDR-tES) intervention treatment or sham treatment condition on Visit 2 before crossing them over to the remaining condition on Visit 4. Each participant will receive a total of 14 days of exposure. Assignments will be counterbalanced among participants. Participants will not be followed up after the conclusion of the eight-week study.

### 2.2. Primary Outcome

We intend to measure the reduction in SOL between sham and stimulation groups using FitBit actigraphy data collected during baseline, treatment, and washout periods.

### 2.3. Secondary Outcomes

We intend to collect subjective survey data for insomnia (ISI) and psychiatric conditions [PHQ-9, PCL-5, STAI], sleep habits [daily sleep diary], HIT, and device experience during experimental sessions. Additionally, we will collect physiological data using actigraphy collected by FitBit Inspire 3 (FitBit Inc., San Franisco, CA, USA) to profile participants’ sleep measures, including their total sleep time (TST) and time spent awake after sleep onset (WASO) and EEG via PeakSleep™ (Teledyne Scientific & Imaging, Durham, NC, USA) to monitor the impact of the stimulation on brain function and activity.

### 2.4. Safety Outcomes

Adverse events (AEs) for this study fall into two different categories. Of note, we believe that the likelihood of these events remains very low.

a.Physical AEs: The final question in our daily sleep diary asks if the participant had any physical or cognitive effects when using the device. If they answer yes, they will be asked to contact study staff and immediately trigger the recording of an AE. Any physical AE will be assessed by study investigators as to their severity, and appropriate medical attention will be provided. Participants with less severe effects will be advised to discontinue the use of the device and treat it at home if possible. For patients with any severe interaction or more serious adverse effects, we will immediately refer them to medical help, and appropriate follow-up will be performed per our stopping criteria.b.Clinical/psychiatric AEs: There may also be a possibility of clinical or psychiatric adverse effects from the device, which will be assessed using the sleep diary, STAI for anxiety, and PHQ-9 for suicidal ideation. Patients are advised to note and report any changes in mood or cognition that occur during the trial.

### 2.5. Eligibility Criteria

Participants are eligible if they are at least 18 to 70 years old, have been diagnosed with sleep-onset insomnia or a self-reported insomnia diagnosis (Insomnia Severity Index score ≥ 15), have sleep onset latency > 30 min 6 months prior to enrollment, and no pharmacologic or non-pharmacologic treatment for insomnia in the last 14 days, including sedating medications at bedtime. Patients who meet any of the exclusion criteria (see Table 1) on the day of consent will be excluded.

After potential participants are selected, the study team will obtain informed consent approved by the USU Institutional Review Board (IRB). Participants will be free to withdraw consent and discontinue participation at any time without prejudice. In addition, the PI may end a participant’s participation in this study for any of the following reasons: (1) if the participant develops any condition that in the judgment of the PI may place them at risks not described above or at greater severity for those described above by continuing his/her participation; (2) if the participant becomes pregnant; (3) if the participant is unable to keep his/her scheduled sessions; (4) if the study is halted by the sponsor, the IRB, or the FDA; and (5) for administrative reasons.

### 2.6. Sample Size

Our trial design was motivated by a power analysis based on the results of a smaller investigational study conducted by Teledyne. A sample size estimation was carried out comparing the mean SOL between the sham and stimulation groups among qualifying patients with a mean sham SOL > 30 min (N = 19, representative subgroup from Teledyne study for this clinical trial). This led to a calculation suggesting that >12 patients per group will enable a comparison across ordered treatment groups for our primary outcome measure.

### 2.7. Recruitment

Participants will be recruited from eligible servicemembers, students, cadets, midshipmen, wounded warriors, and family members from a clinical population at the Walter Reed National Military Medical Center (WRNMMC). Recruitment efforts may include flyers, digital advertisements, email, word of mouth, and direct referrals from a physician. Interested participants will contact the study coordinator via phone or email to schedule an appointment for a screening, and an informed consent visit either at WRNMMC or USU will take place. We will enroll up to 60 patients with clinically diagnosed sleep-onset insomnia with a goal of 48 completing the study (assuming 20% attrition).

## 3. Expected Results

### 3.1. Anticipated Findings

We hypothesize that by applying SDR-tES across the forehead/mastoid for thirty minutes prior to attempting sleep, we can impose slow oscillations within the cortex during wakefulness, and the brain will be more likely to enter a sleep state. Our primary aim is to reduce the time to the onset of sleep by at least 50% from baseline in patients with a >30 min average SOL. Additionally, we will demonstrate that PeakSleep SDR-tES both reduces the time spent awake after sleep onset by 20% and improves the subjective benefits of sleep as measured by the ISI.

### 3.2. Statistical Analyses

Our primary outcome, the change in SOL from baseline as measured via FitBit actigraphy, will be assessed over 14 days of treatment data by constructing a linear mixed effects model accounting for age, baseline SOL, ISI severity, and prior sleep medication use. The changes related to the arm receiving the treatment first will be compared to those achieved by the group receiving the sham first, avoiding any persistent effects of the PeakSleep device. As a secondary outcome, within-subject comparisons between the treatment and sham will also be made using a linear mixed effects model, which will include the order as an additional covariate. Other secondary outcomes will be compared by similarly modeling the objective measures (TST, WASO) and subjective measures obtained at various timepoints across the trial (ISI, PHQ-9). To investigate the differences in sleep measures between treatment groups, we will analyze actigraphy data by evaluating measures of central tendency (the median within a patient and across nights) in each condition (the mean across patients comparing stimulation and sham treatments) imported into MATLAB (R2020). All measures will be investigated as differences from each patient’s individual baselines.

The immediate effects of the stimulation will also be investigated among physiological data, including heart rate and electroencephalography. Changes in heart rate variability and EEG spectral power will be assessed pre- versus post-stimulation, comparing intraindividual changes from baseline between the treatment and sham. For EEG spectral analysis, we will use MATLAB (R2020) to compute spectral power in each frequency band in response to each stimulation event.

The statistical significance of differences between treatment groups for each measure will be tested using either a Wilcoxon Rank Sum test or a paired Student’s *t*-test, depending on whether the data occupy a normal distribution. We will also perform a two-way ANOVA to investigate differences due to sex and the treatment order. A participant’s data will be assessed for inclusion in the study at each visit contingent on obtaining viable data in at least 8 out of the 14 daily sessions during each 2-week period, including viable actigraphy data and the accompanying sleep diary data, successful application of the PeakSleep™ intervention, and adverse events.

## 4. Materials and Equipment

### 4.1. SDR-tES Intervention

Our tES paradigm uses a protocol developed by Teledyne with identical frequency and waveform characteristics for the slow-oscillatory tES paradigm described originally by Marshall and colleagues [16]; only, it is to be applied during wakefulness. The device interfaces wirelessly with a cell phone over low-energy Bluetooth and is controlled from an easy-to-use application.

#### 4.1.1. Stimulation Frequency

We will apply the stimulation over 2 sets of anode/cathode pairs. Anodes are located approximately at the frontopolar 1 (Fp1) and frontopolar 2 (Fp2) locations, as measured in the standard 10–20 EEG electrode system. Cathodes are placed on the ipsilateral mastoids. We will use a pulsed direct current waveform at a frequency of current oscillation at 0.75 Hz.

#### 4.1.2. Stimulation Protocol

PeakSleep™ is a constant current device that delivers a stable stimulation as a function of the impedance measured across the two sets of medical grade hydrogel electrodes (Axelgaard Little PALS; Axelgaard Manufacturing, Fallbrook, CA, USA), which achieve skin impedances in the 20–50 kOhm range (a DC impedance measurement) and carry low sensation, typical of other tES devices. The device gradually increases its peak stimulation current, limited to a range between 100 and 500 A per electrode. Devices are configured to deliver 100 stimulation trains over 30 min, where each train has 6 pulses of a 0.75 Hz trapezoidal stimulation. The inter-train interval is 10 s, leading to a total stimulation time of <14 min with a maximum dose of 1 mA (500 μA per electrode pair). We will limit the total stimulation time to 30 min per day at the prescribed intensities in accordance with the recommended best practices. Our selected stimulation protocol falls within the appropriate range of current published guidelines on best practices for the safe application of tES in most cases [14].

### 4.2. Sham Condition

Devices will be configured to deliver a low amplitude (e.g., 100 μA) waveform of a different frequency (e.g., 25 Hz) for the same treatment duration. The purpose of the sham stimulus is to control for any placebo effect arising from the sensation of the stimulation. Beyond these differences in the frequency and amplitude of stimulation, devices will be operated in exactly the same way during the sham treatment. Participants and investigators will both be blinded as to their assignment, and only the study monitor will know the assignment of treatment conditions.

### 4.3. Device Safety

The devices are contained in individual pelican cases that are easy to store in a secure manner. Participants will be advised not to share the device and to secure it when not actively in use. Additionally, the device has a safety check that prohibits more than 40 min of use in a given 24 h period. Therefore, if others are using the device in addition to the patient, the patient will know about it and will be instructed to contact the research staff.

## 5. Detailed Procedure

### 5.1. Visit 1 (Enrollment)

If the participant meets all inclusion criteria, informed consent will take place prior to any experimental procedures. Once consent is given and participants understand the full scope of the study procedures, they will be asked to complete a demographics questionnaire, the patient health questionnaire (PHQ-9), the Posttraumatic Stress Disorder (PTSD) Checklist for DSM-5 (PCL-5), the State Trait Anxiety Inventory (STAI) and the Headache Impact Test (HIT). Suicidal ideation is one of the exclusion criteria for further participation indicated by item 9 on the PHQ-9. Any participant responding to this item with a score > 0 will be interviewed by the PI, and if further medical attention is deemed necessary, the participant will be dismissed from any further participation and referred to the appropriate medical help. If a participant fails to meet the inclusion/exclusion criteria for the study, then all data from screening materials (PHQ-9, PCL-5, STAI, HIT, ISI) will be deleted. Participants will be offered compensation of up to USD 1000 for the completion of the study, with an opportunity to receive compensation at the completion of each in-person visit. A detailed schedule of assessments to be collected during the 8-week study can be found in Table 2.

#### 5.1.1. Actigraphy Device Training

The participant will be trained on the device and asked to wear the actigraphy device (FitBit) continuously for the next 2 weeks. The first two weeks of the study involve collecting baseline sleep assessment data with a particular focus on identifying a pre-intervention SOL, which is the primary outcome.

#### 5.1.2. Subjective Questionnaire Training

Participants will then be informed of when and how to complete the required daily sleep diary for the next two weeks. Cumulatively, Visit 1 may take up to 1 h.

### 5.2. Visit 2 (Treatment 1)

Participants are block-randomized to treatment conditions based on the order (sham or stimulation first) and will be balanced with respect to gender. Participants who successfully complete 2 weeks of baseline collection will be enrolled in the first treatment group of their assigned order. To assess their insomnia baseline, they will complete the Insomnia Severity Index (ISI) during this visit. They will be trained on the use of the PeakSleep™ device and sent home for 2 weeks. The PeakSleep™ device will deliver therapy and collect EEG data for the investigation of neural activity for both stimulation and sham conditions. The participant’s response to the first treatment condition will also be measured with actigraphy and a daily sleep diary completed on a phone app, which also controls the PeakSleep™ device.

#### Initial Headset Training Session

The individual will participate in a training session on how to use the headset device. Participants will complete a short (5 min) trial session to demonstrate that they know how to use the device and to monitor any short-term adverse reactions to the stimulation prior to home use. Visit 2 is expected to take about 30 min.

### 5.3. Visit 3 (Washout)

During Visit 3, we will collect PeakSleep™ devices and administer a second ISI to assess participants’ response to treatment condition 1. Participants will also be requested to fill out a short questionnaire asking about the occurrence and details of any adverse events during the past two weeks with the device, which the investigator will review during the visit. Participants will complete another PHQ-9 to reassess any impact on depression or suicidal ideations. Participants will return home without the PeakSleep™ device but will continue to wear the actigraphy devices and complete the daily sleep diary in order to monitor sleep habits during a 2-week washout period. The purpose of the washout interval is to investigate any lingering effects from the first intervention and to ensure that those receiving the stimulation first return to a proper baseline prior to the sham treatment. Visit 3 will take approximately 1 h.

### 5.4. Visit 4 (Treatment 2)

Participants who were assigned to the stimulation treatment group during Visit 2 will be assigned to the sham group and vice versa. Participants will take the ISI once again for a second baseline measurement and will be sent home for the final 2 weeks to collect data on their responses to the second treatment condition. Their response to the second treatment condition will be measured with actigraphy and a daily sleep diary. EEG data will also be collected again from PeakSleep™ for the investigation of neural activity. Visit 4 will take approximately 30 min.

### 5.5. Visit 5 (Study Exit)

Participants will return the actigraphy and PeakSleep™ devices. We will administer a final measurement of the ISI to assess the impact of the second treatment condition over the previous 2 weeks. Participants will again fill out the PHQ-9 to reassess any impact on depression or suicidal ideations. Participants will repeat the adverse event questionnaire and be asked to complete a questionnaire on their user experience with the device. Visit 5 will take approximately 1 h.

## 6. Discussion

### 6.1. Strengths and Limitations

There remains ample evidence to support the need for effective interventions to improve sleep capabilities in the context of performing high-operational demands and their long-term sequelae on our active duty and veteran populations. Pilot data support Teledyne’s PeakSleep™ device as an attractive and feasible treatment for individuals with sleep-onset insomnia, thus warranting its replication and validation in a larger military population. This study will be the first to investigate the effects of PeakSleep™ in our military population by specifically analyzing sleep onset and duration, raw EEG data, and subjective measures of sleep.

Notable strengths of this study are the repeated measurement design, the device’s simplicity and safety profile, and an expansive recruitment pool. A repeated measurement study design enhances statistical power by controlling differentiating factors between participants, given that our participants act as their own control and allow for fewer participants to detect a desired effect. Additionally, PeakSleep™ is a simple and safe device with straightforward instructions that maximize patient adherence to the protocol and encourage the use of the device. Lastly, this study will take advantage of the military health system’s unified charting software and the extensive patient population present at the WRNMMC. The study population will largely include active-duty service members who are WRNMMC Sleep Disorders Center patients and have been diagnosed with clinical insomnia, which will potentially streamline a source of participants.

Important limitations of this study include the study’s small sample size, though it is larger than Teledyne’s pilot study, the expected minor proportion of civilian participants recruited, a lack of long-term follow-up, and a reliance on Fitbit actigraphy data. Although we are able to use FitBit sleep phase data for the purpose of this study, given our comparison across the same device and participant, we recognize its limited reliability in deep sleep measures compared to PSG. In future trials, investigators should include more advanced forms of actigraphy monitoring, such as a Motionlogger Watch or Actiwatch, or include a pre-/post-treatment PSG with spectral analysis, given FitBit’s limited specificity in sleep staging [26,27]. In order to maximize the generalization of our results to a civilian population, additional studies should include the targeted recruitment of the civilian population to further characterize the potential effects of tES and generate access to a larger patient population. Lastly, we recognize that any long-term effects have yet to be elucidated, and future studies should investigate longer treatment periods and include follow-up data points.

### 6.2. Potential Implications

Despite previous trials demonstrating the effectiveness of tES in improving deep sleep and memory consolidation, there remains a gap in the implications of tES in patients, military and civilian alike. We aim to develop and demonstrate a new treatment option for those with insomnia that is effective, safe and exhibits a favorable side effect profile compared to current treatment options. If successful in addressing sleep onset and improving the subjective measures of sleep, PeakSleep could be expanded as an intervention for various sleep disorders ranging from chronic insomnia to insufficient sleep syndrome. Future studies should shift the focus to specifically investigate PeakSleep’s ability to enhance SWS. Addressing both sleep onset and the maintenance of SWS could have enormous implications for civilian and military sectors and, most importantly, warfighters in whom efficient sleep onset may be critical for operational success.

## Figures and Tables

**Figure 1 clockssleep-07-00003-f001:**
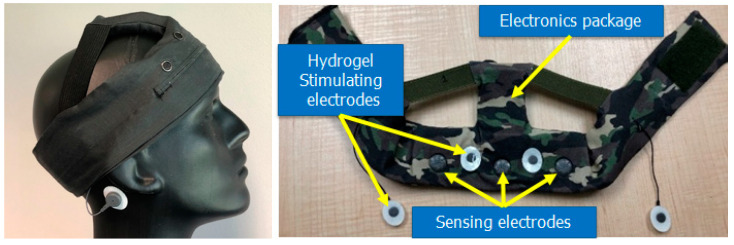
PeakSleep™ senses brain activity using three EEG sensors on the forehead and delivers a current through two pairs of hydrogel electrodes located on the forehead and matched to ipsilateral mastoid locations.

**Table 1 clockssleep-07-00003-t001:** Exclusion criteria.

Exclusion Criteria:
1. Neurologic conditions that include seizures or increase their risk, including recent, multiple, or severe concussion or traumatic brain injury, stroke, multiple sclerosis, or cognitive impairment. 2. Any psychiatric disorder requiring weekly or more frequent clinical monitoring or medication changes in the last 4 weeks. 3. A history of neurodevelopmental disorders such as attention deficit hyperactivity disorder, a learning disability, or developmental delay. 4. Any inpatient hospitalization or major surgery or medical procedure within the past 6 months. 5. Hearing impairments requiring implanted or external devices for amplification. 6. Current substance use disorder within the past year, not including nicotine. 7. The current use of narcotics or opioid-based medications for the treatment of pain with or without a prescription within the last year. 8. Change in psychotropic (non-sleep-related) medications within the last 4 weeks. 9. Consuming more than 10 alcoholic beverages per week. 10. Treatment of drug or alcohol use/abuse within the past 1 year. 11. Sleep disorders that require treatment other than insomnia. 12. Any motor coordination deficits that interfere with the use of the tES device. 13. Participants should not have trauma/cuts/rashes on their forehead or behind the ears that would interfere with wearing the device or cause discomfort for the research subject. 14. Tattoos on the head. 15. Non-removable metal anywhere in the body except bridges or fillings, including pacemakers, defibrillators, cochlear implants, brain implants including deep brain stimulators, or other implanted devices. 16. Any suicidal attempts within the last 12 months. 17. Any other condition that the investigator believes would prevent the completion of the study or put the participant at risk. 18. Any suicidal ideations or thoughts of self-harm within the last 2 weeks. 19. Pregnant or believes there is a chance of pregnancy. ^1^

^1^ Safety data for tES use in pregnant women are scarce, and we will exclude this population if the questionnaire reveals they are pregnant or are trying to become pregnant. In the event that the patient is untruthful regarding their pregnancy status, we view the risk as insignificant given the reported studies and the fact that our dose is less than one-fourth of the dose used in such studies.

**Table 2 clockssleep-07-00003-t002:** Detailed schedule of assessments during 8-week data collection period.

	Timepoint				
Assessment	Visit 1	Visit 2	Visit 3	Visit 4	Visit 5
Informed Consent	X				
Patient Health Questionnaire	X		X		X
Posttraumatic Stress Disorder Checklist for DSM-5	X				
State Trait Anxiety Inventory	X				
Headache Impact Test	X				
Insomnia Severity Index		X	X	X	X
Daily Sleep Diary	X	X	X	X	X
Adverse Events Questionnaire			X		X
Device Experience Questionnaire					X
FitBit Actigraphy Data		X	X	X	X
PeakSleep™ EEG Data			X		X

## Data Availability

Clinical data will be collected from AHLTA and MHS Genesis electronic medical records and de-identified using an alphanumeric code. These data will be shared with research staff. The wearable study results will be disseminated via publications, national meetings/conferences, and other means of communication to both scientific and lay audiences.

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
