# Peer review of "Wearable Neurotechnology for the Treatment of Insomnia: The Study Protocol of a Prospective, Placebo-Controlled, Double-Blind, Crossover Clinical Trial of a Transcranial Electrical Stimulation Device"

_2624-5175, 2025, doi:10.3390/clockssleep7010003_

Round 1

Reviewer 1 Report

Comments and Suggestions for Authors

This protocol investigates whether a translational device, Teledyne PeakSleep, can reduce sleep onset latency and awake time after sleep onset. Protocol is well designed, and safety concerns are low. The study could make a significant contribution to the field. However, there some issues need be clarified. My comments is appended.

General comment

1. Abbreviation should be defined in the first appearance.

   Line 45 VA.

Line 88 SDR.

Introduction

2. In the phas1 study in 2019-2020, PeakSleep had been confirmed can improve sleep quality. It is import to report the reason of performing the study. Whether the function of the device has some improvement? Whether the results in previous study did not meet the expectation of the sponsor?

3. It is important to report the advantages of Peaksleep compared to other tES instruments?

Methods

4. How to define the device is effective? In primary outcome section, only collecting actigraphy data is described, but not how to define whether the device is effective.

5. It is better use a figure to show study procedure.

6. PeakSleep consists of a soft fabric headband with integrated electronics for dry EEG sensing and electrical stimulation.

Participants were asked to use the device during sleep. How to prevent this device

from interfering with a person's sleep?

Discussion

7. In comparison to phase 1 study, What is the strength of the study? 

Author Response

Comments 1: Abbreviation should be defined in the first appearance. Line 45- VA and Line 88- SDR.

Response 1: VA abbreviation corrected on page 2, paragraph 1, line 45. SDR abbreviation corrected on page 2, paragraph 5, line 88. 

Comments 2: In the phase 1 study in 2019-2020, PeakSleep had been confirmed can improve sleep quality. It is important to report the reason of performing the study. Whether the function of the device has some improvement? Whether the results in previous study did not meet the expectation of the sponsor?

Respnse 2: Thank you for your comment. Within our introduction section we discussed that Teledyne first developed PeakSleep as an alternative to the standard pharmacologic and behavioral interventions for individuals with insomnia. Their internal study successfully demonstrated that PeakSleep can reduce sleep onset latency in civilians with insomnia (page 2, paragraph 3, lines 63-65). Teledyne were pleased with their results and were open to partner with an independent investigator for follow-on trials.

Comments 3: It is important to report the advantages of PeakSleep compared to other tES instruments?

Response 3: Agreed. We reference several studies that have investigated transcutaneous electrical stimulation however Teledyne’s device was the first to demonstrate in-human evidence of a portable sleep priming device. As a result, we elected to withhold from explicitly comparing PeakSleep to other TES or TMS devices available on the market (page 2, paragraph 2, lines 56-59). Nonetheless, by combining comfort, ease of use, cost-effectiveness, and clinical evidence of improving sleep, PeakSleep could offer a comprehensive and practical advantage over other sleep aids/wearable neurotechnology on the market.

Comment 4: How to define the device is effective? In primary outcome section, only collecting actigraphy data is described, but not how to define whether the device is effective.

Response 4: Thank you for your comment. We describe the primary outcome as a decrease in sleep onset latency (page 3, paragraph 2, lines 95-96).

Comment 5: It is better to use a figure to show study procedure.

Response 5: Thank you for pointing this out. Given the various modalities of data collected during and in-between in-person visits, we opted to construct a table of scheduled assessments (Table 2) for simplicity rather than design a figure for the protocol design manuscript. We can certainly construct a figure of study procedure when we complete the study and publish our final results.

Comment 6: PeakSleep consists of a soft fabric headband with integrated electronics for dry EEG sensing and electrical stimulation. Participants were asked to use the device during sleep. How do you prevent this device from interfering with a person's sleep?

Response 6: Thank you for bringing up this concern. The training session during Visit 2 will allow participants to familiarize themselves with the device and troubleshoot any issues with the study team to minimize any discomfort that could interfere with sleep. Its noninvasive and comfortable headband design is specifically tailored for use during sleep, emphasizing minimal disruption to daily routines. Although the phase 1 trial completed by Teledyne did not report the device interfered with their participant’s sleep, our participants will also complete a device experience questionnaire to assess for impact on sleep and inform any necessary changes for future studies.

Comment 7: In comparison to phase 1 study, what is the strength of this study? 

Response 7: Thank you for your comment. We have partnered with Teledyne after their successful phase 1 trial to expand upon their findings by testing their device in our unique military population, with a larger sample size, and by also collecting additional physiologic and subjective sleep measures (page 2, paragraph 4, lines 72-79).

Reviewer 2 Report

Comments and Suggestions for Authors

Clocks Sleep 3248875

This is a description of a proposed industry-sponsored, single-center RCT of a device (PeakSleep) to treat insomnia in a specific target population of military personnel.   The device uses transcranial electrical stimulation to target frontal brain. Preliminary data is a single-arm trial. Design is randomized cross-over. Proposed primary outcome is FitBit SOL.

My major critique is their selection of an SOL regression as the primary outcome.

1.        Please state whether this an early phase or definitive phase trial? 

2.        It took some reading to find out what the primary outcome of the trial is, and I believe it’s SOL

a.        However, the protocol in Methods section 4 emphasizes collection of ISI. Why, if SOL is the primary outcome. Id get the primary outcome clearly stated and early.

b.        Please defend the use of SOL rather than ISI as a primary outcome measure. I would only consider a treatment of insomnia effective if it decreases ISI; SOL/TST is usually a secondary outcome in insomnia trials. Even if it’s an early phase trial, ISI is the standard.

c.        Accordingly, I’d recommend emphasizing ISI change scores as the primary outcome, and power the study appropriate for that change score. 

d.        How is SOL calculated from FitBit: what is the criteria? Is this embedded in FitBit interpretational software, or a threshold measure selected by authors. Will a sleep diary supplement actigraphy? Usually, actigraphy is noisy enough that some filtering/adjustments with sleep diaries are necessary to extract useful information from actigraphy. I’d be wary about accepting any conclusions from a sealed, proprietary algorithm. 

e.        In summary, I’d like to see an ISI range declared as a “go/no-go” criterion for moving forward. I don’t approve of the SOL regression approach with various corrections as either an early-phase or Phase III trial for insomnia. Either it works, or it doesn’t. I think this is a hard line for acceptance of this trial.

3.        Back to protocol stage: is this trial intended to lead to FDA approval? Is it covered by an IDE?

4.        Inclusion/exclusion: epilepsy/seizures an exclusion? Probably not a significant exclusion in a military cohort, but I’d be worried that a seizure would effectively kill this device in early trial.

Author Response

Comment 1: Please state whether this an early phase or definitive phase trial? 

Response 1: Thank you for pointing out this omission. Added clarification of phase on page 2, paragraph 5, line 83. 

Comment 2a: However, the protocol in Methods section 4 emphasizes collection of ISI. Why, if SOL is the primary outcome. I’d get the primary outcome clearly stated and early.

Response 2a: Agreed. The primary and secondary outcomes were re-arranged and are now included within the Experimental Design Section 2 (page 3, paragraphs 2 and 3, lines 95-103). Additional wording was also added within the Methods section 4 to clarify that baseline data will be collected between Visit 1 and 2, with a particular focus on establishing a baseline SOL, the primary outcome (page 6, paragraphs 5 and 6, lines 209-215). 

Response 2b: Please defend the use of SOL rather than ISI as a primary outcome measure. I would only consider a treatment of insomnia effective if it decreases ISI; SOL/TST is usually a secondary outcome in insomnia trials. Even if it’s an early phase trial, ISI is the standard.

Comment 2b: Thank you for your comment. Our approach focuses on the use of short duration repetitive - transcranial electrical stimulation to enhance brain rhythms present during the transition from wake to sleep and hypothetically prime the brain prior to attempted sleep onset to facilitate the transition to sleep. Given the hypothesizing mechanism by which PeakSleep will treat insomnia, we are more interested in a measure that directly measures the time to fall asleep rather than a self-report questionnaire measuring the severity of insomnia symptoms. Although ISI is often used as a primary outcome for trials investigating insomnia, SOL provides a quantifiable and objective measure of a treatment’s impact, especially when used in conjunction with sleep diaries. ISI captures a participant’s subjective experience of an intervention on their perceived insomnia severity and functional impairment, which is valuable and why we are also collecting this data throughout the study. However, we are primarily interested in the effects of PeakSleep on specific physiological sleep patterns and parameters. Additionally, Teledyne’s initial trial demonstrated a reduction in SOL and our study is building off of theirs by expanding sample size and narrowing the diagnostic criteria to sleep onset insomnia.

Comment 2c: Accordingly, I’d recommend emphasizing ISI change scores as the primary outcome, and power the study appropriate for that change score. 

Response 2c: Please see "Response 2b" for addressing this comment. 

Comment 2d: How is SOL calculated from FitBit: what is the criteria? Is this embedded in FitBit interpretational software, or a threshold measure selected by authors. Will a sleep diary supplement actigraphy? Usually, actigraphy is noisy enough that some filtering/adjustments with sleep diaries are necessary to extract useful information from actigraphy. I’d be wary about accepting any conclusions from a sealed, proprietary algorithm. 

Response 2d: Thank you for bringing up this concern. We agree we must pay particular attention to our data analysis when using a sealed, proprietary algorithm (that interprets data from an accelerometer and heart rate variability) as the tool for determining sleep onset. We also recognize FitBit is a significant limitation in our study design for actigraphy when compared to other actigraphy monitoring options, but funding was a limiting factor. With that being said, several studies have demonstrated FitBit can be comparable to PSG when measuring SOL and identifying sleep-wake states despite over or underestimation of other sleep measures to include WASO, TST, or SE. The studies agree FitBit cannot replace PSG though there are arguably acceptable situations where they can be applied as substitutes given the promising comparisons to PSG. For this reason, we feel confident in accepting SOL collected via FitBit actigraphy for our Phase II trial with plans to expand to research grade devices for future studies (page 9, paragraph 5, lines 318-319). Additionally, as you mentioned, participants will be completing a daily sleep diary for the entire 8-weeks via the PeakSleep app to supplement our actigraphy (page 3, paragraph 3 line 99 and page 6, pagraph 6, lines 213-214) and maximize accuracy for calculating SOL. Please see the additional below referrences. 

Haghayegh S, Khoshnevis S, Smolensky MH, Diller KR, Castriotta RJ. Accuracy of Wristband Fitbit Models in Assessing Sleep: Systematic Review and Meta-Analysis. J Med Internet Res. 2019 Nov 28;21(11):e16273. doi: 10.2196/16273. PMID: 31778122; PMCID: PMC6908975.

Dong X, Yang S, Guo Y, Lv P, Wang M, Li Y. Validation of Fitbit Charge 4 for assessing sleep in Chinese patients with chronic insomnia: A comparison against polysomnography and actigraphy. PLoS One. 2022 Oct 18;17(10):e0275287. doi: 10.1371/journal.pone.0275287. PMID: 36256631; PMCID: PMC9578631.

https://publications.ersnet.org/content/erj/60/suppl66/3475

Comment 2e: In summary, I’d like to see an ISI range declared as a “go/no-go” criterion for moving forward. I don’t approve of the SOL regression approach with various corrections as either an early-phase or Phase III trial for insomnia. Either it works, or it doesn’t. I think this is a hard line for acceptance of this trial.

Response 2e: Thank you for your comment. Although we disagree on the primary outcome and the critical nature of ISI for this specific study as a "go/no-go" component, tracking changes in ISI baseline after the intervention will be an valuable secondary outcome.

Comment 3: Back to protocol stage: is this trial intended to lead to FDA approval? Is it covered by an IDE?

Response 3: Thank you for requesting this clarification. The FDA has already designated PeakSleep as non-significant risk and does not require an IDE for this study. 

Comment 4: Inclusion/exclusion: epilepsy/seizures an exclusion? Probably not a significant exclusion in a military cohort, but I’d be worried that a seizure would effectively kill this device in early trial.

Response 4: Thank you for requesting a clarification on exclusion criteria. An exclusion table was added to the manuscript (page 4, table 1). 

Round 2

Reviewer 2 Report

Comments and Suggestions for Authors

The authors addressed my comments appropriately. I still think ISI is the better measure in the clinical assessment of "insomnia" and I'd reject this study if insomnia were the target disorder. However, in my re-reading of the protocol and the authors' responses, since the cohort is military and insufficient sleep syndrome, rather than insomnia, is one of the target disorders, I think SOL in this context is an acceptable primary outcome. Thus, in section 6.3, the authors should state that insufficient sleep syndrome as well as chronic insomnia are target disorders. 

Author Response

Comments 1: The authors addressed my comments appropriately. I still think ISI is the better measure in the clinical assessment of "insomnia" and I'd reject this study if insomnia were the target disorder. However, in my re-reading of the protocol and the authors' responses, since the cohort is military and insufficient sleep syndrome, rather than insomnia, is one of the target disorders, I think SOL in this context is an acceptable primary outcome. Thus, in section 6.3, the authors should state that insufficient sleep syndrome as well as chronic insomnia are target disorders. 

Response 1: Thank you for your swift reply and review of our previous response. We agree that multiple sleep disorders could be studied within our protocol but elected to focus on sleep onset insomnia given this population was chosen by the pilot study. However, it is important to clarify we hope PeakSleep will be expanded as an intervention for any multiple sleep disorders and have added wording discussing the future implication of PeakSleep's therapeutic potential for sleep disorders outside of sleep onset insomina and specifically included insufficient sleep syndrome (page 10, paragraph 1, lines 332-334).